# Development of a Rapid and Sensitive Fluorescence Sensing Method for the Detection of Acetaldehyde in Alcoholic Beverages

**DOI:** 10.3390/foods11213450

**Published:** 2022-10-31

**Authors:** Yisong Liu, Chunfeng Liu, Xin Xu, Chengtuo Niu, Jinjing Wang, Feiyun Zheng, Qi Li

**Affiliations:** 1Key Laboratory of Industrial Biotechnology, Ministry of Education, School of Biotechnology, Jiangnan University, Wuxi 214122, China; 2Laboratory of Brewing Science and Technology, School of Biotechnology, Jiangnan University, Wuxi 214122, China

**Keywords:** acetaldehyde, fluorescence detection, fluorescent probe, alcoholic beverages, photoinduced electron transfer

## Abstract

Acetaldehyde is regarded as an important flavor compound in alcoholic beverages. With the advantages of rapidity, low cost and high sensitivity, fluorescent probe could be used as a new tool for the detection of acetaldehyde. Here, an effective fluorescence sensing method based on fluorescent probe N1 (FPN1) was established in this study. The function of FPN1 relies on the nucleophile substitution reaction and photoinduced electron transfer (PET), resulting in a fluorescence increase. Remarkably, the pretreatment background removal method (BRM) was successfully applied for removal of the interference of pyruvate and acetal. The linearity range (LR), limit of detection (LOD) and recovery of the fluorescence sensing method with BRM were 0.0053–200 mg/L, 0.0016 mg/L and 94.02–108.12%, respectively, which showed a broader detection range and better performance on sensitivity compared with the traditional quantitation using gas chromatography (GC). Furthermore, successful application of the method in real samples indicated the advantages of low-cost and rapidity for small-scale detection while assuring the accuracy, which provides a new strategy for the detection of acetaldehyde concentration in alcoholic beverages.

## 1. Introduction

For alcoholic beverages, high alcohols, esters, carbonyls and sulfides have great impact on flavor and stability [1]. Aldehydes are the main carbonyl compounds in alcoholic beverages. It is worth noting that acetaldehyde accounts for about 60–97% (approximately over 60% for beer, over 75% for liquor, over 70% for *Huangjiu* and over 90% for wine) of the aldehydes [2,3,4,5,6,7,8]. Therefore, acetaldehyde is regarded as one of the most important flavor components (general concentration range is approximately 1–30 mg/L for beer, 40–500 mg/L for liquor, 30–140 mg/L for *Huangjiu* and 30–130 mg/L for wine). In alcoholic beverages, acetaldehyde is mainly produced during fermentation by decarboxylation of pyruvate by pyruvate decarboxylase. In general, acetaldehyde concentration is expected to be low to avoid the bad flavor caused by acetaldehyde [9], and there are several factors which could be managed in order to control acetaldehyde concentration, such as the characteristics of yeast, oxygen content, pH, raw materials and temperature. Moreover, acetaldehyde is possibly carcinogenic to humans, and was considered a class one carcinogen by the International Agency for Research on Cancer in 2009 [10]. Thus, the measurement of acetaldehyde in alcoholic beverages is an essential aspect of quality control and food safety.

Until now, there has been a certain series of methods for the detection of acetaldehyde in alcoholic beverages, including gas chromatography (GC) and high performance liquid chromatography (HPLC). Acetaldehyde can be detected in mixtures in trace concentrations with these analytical measurements, which have a high selectivity. In order to improve the accuracy, the solid phase microextraction or derivatization technology is usually combined with the GC and HPLC method [11,12]. However, GC and HPLC have a significant challenge in dissecting the disadvantages of time-consuming, costly, low sensitivity and complex procedures. In contrast, several detection methods based primarily on fluorescent probes have been developed for application in cell imaging, environmental analysis, food science and the medicine industry in recent years [13,14,15], which shows the advantages of simplicity, low cost and high efficiency. In alcoholic beverages, some probes have been implemented, which provides insight into the design strategies and applications [16,17,18,19,20]. In particular, some studies have proposed that the concentration of acetaldehyde in alcoholic beverages can be quantified by fluorescence detection of NADH, which is consumed by reverse reaction of acetaldehyde dehydrogenase and provides a key link with our work [21,22,23,24,25], although the design principle is different from the general principles for fluorescent probes [16].

Therefore, in order to explore a novel, rapid, sensitive and low-cost method based on fluorescent probes for detecting acetaldehyde in alcoholic beverages, here, we developed a fluorescence sensing method based on fluorescent probe N1 (FPN1), which could respond with acetaldehyde by the photoinduced electron transfer (PET) principle. The fourth position substituent of the fluorescent group (benzoxadiazole) was designed as a nitro group, which strongly withdraws electrons, and a reactive group was designed to replace the seventh position substituent with a hydrazine group. Acetaldehyde competes for the electron transferred from the hydrazine group to the nitro group after the reactive group is combined with acetaldehyde, which results in the disruption of PET and enhancement of the fluorescence intensity [26]. Factors such as pH, solvent, probe concentration and response time were considered in the optimization for the response between FPN1 and acetaldehyde. It is worth noting that we are the first to use the general principle PET of the fluorescence probe design strategies to detect acetaldehyde in alcoholic beverages.

More importantly, in order to remove the interference of pyruvate and acetal from alcoholic beverages, we evaluated the effectiveness of the background removal method (BRM) through anti-interference analysis. This is the first time that BRM has been proposed in the application of fluorescent probe detection. In order to demonstrate the advantages of detection, the performance and the utility of the fluorescence sensing method with BRM were demonstrated and compared with the traditional detective method. With the advantages of broader detection range, better performance on sensitivity and efficiency compared with the traditional quantitation using GC, the fluorescence sensing method based on FPN1 with BRM is significant for monitoring and controlling the product quality in alcoholic beverages.

## 2. Materials and Methods

### 2.1. Chemicals and Apparatus

The chemicals dimethyl sulfoxide (DMSO, 99%), hydrazine hydrate (N_2_H_4_·H_2_O, 99%), methanol (CH_3_OH, 99%), trichloromethane (CHCl_3_, 99%), dichloromethane (CH_2_Cl_2_, 99%), N,N-dimethylformamide (DMF) (C_3_H_7_NO, 99%), hydrochloric acid (HCl, 38%) and HPLC grade chloroform acetonitrile (CH_3_CN, 99.9%) were obtained from Sinopharm (Sinopharm, Shanghai, China). The reagents 4-chloro-7-nitro-1,2,3-benzoxadiazole (NBD-Cl, 98%), acetaldehyde (CH_3_CHO, 99.5%), acetal (C_6_H_14_O_2_, 99.5%), furfural (C_5_H_4_O_2_, 99.5%), 5-hydroxymethylfurfural (5-HMF) (C_6_H_6_O_3,_ 99.5%), acetoin (C_4_H_8_O_2_, 97%), 2,3-butanedione (C_4_H_6_O_2_, 99%), 2,3-pentanedione (C_5_H_8_O_2_, 99%), 2,3-hexanedione (C_6_H_10_O_2_, 99%), pyruvate (C_3_H_4_O_3_, 99%), propanone (CH_3_COCH_3_, 99%), hydroxypropanone (C_3_H_6_O_2_, 99%), α-ketoglutaric acid (C_5_H_6_O_5_, 99%), methylglyoxal (C_3_H_4_O_2_, 99%), propionaldehyde (C_3_H_6_O, 99%), butyraldehyde (C_4_H_8_O, 99%), isobutyraldehyde (C_4_H_8_O, 99.5%), isovaleraldehyde (C_5_H_10_O, 99.5%), caproaldehyde (C_6_H_12_O, 99.5%), nonanal (C_9_H_18_O, 99.5%), heptanaldehyde (C_7_H_14_O, 99.5%), octanal (C_8_H_16_O, 99.5%), benzaldehyde (C_7_H_6_O, 99%), phenylacetaldehyde (C_8_H_8_O, 99%), methylglyoxal (C_3_H_4_O_2_, 99%), glyoxal (C_2_H_2_O_2_, 40%), L-asparagine (Asn, 99%), L-glutamine acid (Gln, 99%), ethanol (C_2_H_5_OH, 99.5%), 1-propanol (C_3_H_8_O, 99%), 1-butanol (C_2_H_6_O, 99%), isobutanol (C_4_H_10_O, 99%), isoamyl alcohol (C_5_H_12_O, 99%), 3-methylbutanol (C_5_H_12_O, 99%), β-phenylethanol (C_8_H_10_O, 98%), acetic acid (CH_3_COOH, 99.7%), lactic acid (C_3_H_6_O_3_, 98%), caproic acid (C_6_H_12_O_2_, 98%), ethyl acetate (C_4_H_8_O_2_, 99.5%), isopentyl acetate (C_7_H_14_O_2_, 99.5%), ethyl caproate (C_8_H_16_O_2_, 99.5%), ethyl lactate (C_5_H_10_O_3_, 99.5%), monomethyl succinate (MES) (C_5_H_8_O_3_, 98%) and tetramethylpyrazine (TMP) (C_8_H_12_N_2_, 98%) were purchased from Aladdin (Aladdin, Shanghai, China). An organic silicon defoamer was purchased from Sinopharm (Sinopharm, Shanghai, China).

An HPLC spectrum was performed on Chromaster CM5110 (Hitachi, Japan). High resolution mass spectroscopy (HRMS) was performed using Agilent HPLC System/6110 MS (NYSE: A, Palo Alto, CA, USA). NMR spectra were performed on a AVANCE III 500 MHz (Bruker, German). An IR spectrum was performed on ThermoFisher Nicolet (ThermoFisher, Waltham, MA, USA). Fluorescence spectra were recorded using a Biotek synergy H4 fluorescence spectrometer with a temperature controller (Bio Tek, VT, USA). Determination of acetaldehyde by GC method was measured with the GC-2010 PerkinElmer TurboMatrix 16 with flame ionization detection (FID) connected to a HS-10 headspace sampler (Shimadzu, Japan).

### 2.2. Synthesis of FPN1

FPN1 was synthesized according to the previously published studies [27]. The synthetic route of FPN1 is displayed in Figure 1. NBD-Cl (compound 1; 500 mg) was dissolved in chloroform (50 mL). The mixture was added to hydrazine hydrate (50 mL, 5%) in the flask, slowly, while stirring. The mixture was stirred at room temperature for 6 h. The precipitate was collected by suction filtration and washed with dichloromethane, resulting in obtaining the FPN1 (compound 2; 461.6 mg) as a yellow-brown solid with a yield of 91.2%. The structure of FPN1 was confirmed using HPLC, HRMS, ^1^H NMR and IR.

### 2.3. Optimization of Detection Conditions between Acetaldehyde and FPN1

The reactions of FPN1 (300 mg/L) with and without acetaldehyde (10 mg/L) at different pH levels (1–12) show the pH sensitivity. In order to select an appropriate solvent, FPN1 (300 mg/L) and acetaldehyde (10 mg/L) were reacted in DMSO, acetonitrile, ethanol, DMF, PBS and water. In order to determine the optimal concentration of probe solution, acetaldehyde (10 mg/L) was reacted with different concentrations of FPN1 (100, 200, 300, 400, 500, 600, 700, 800 mg/L) in acetonitrile. All of the fluorescence intensity values were recorded immediately after the solutions were well mixed.

In order to identify the optimal emission wavelength, the fluorescence spectral of FPN1 (300 mg/L) in the presence of acetaldehyde (2, 5, 10, 20, 50, 100 mg/L) was recorded. In order to determine the photosensitivity of the reaction, recording the fluorescence intensity of FPN1 (300 mg/L) in the presence of acetaldehyde (100 mg/L) with or without light every five minutes. In order to determine the optimal reaction time, time-dependent fluorescence intensity changes of FPN1 (300 mg/L) in the presence of acetaldehyde (2, 5, 10, 20, 50, 100 mg/L) were recorded every five minutes.

### 2.4. Preparation of FPN1 Solution

The FPN1 solution contained acid solution and probe solution. HCl (4 mL) was diluted in acetonitrile (50 mL) as the acid solution. FPN1 (6 mg) was dissolved in the acetonitrile and DMSO solution (10:1, *v/v*, 20 mL) as the probe solution. Both of the two solutions were stored at room temperature and used immediately.

### 2.5. Analytes of the Selectivity Analysis

In order to show the selectivity of FPN1, there were 39 analytes considered in total. Acetaldehyde, acetal, furfural, 5-HMF, acetoin, 2,3-butanedione, 2,3-pentanedione, 2,3-hexanedione, pyruvate, propanone, hydroxypropanone, α-ketoglntaric acid, methylglyoxal, propionaldehyde, butyraldehyde, isobutyraldehyde, isovaleraldehyde, caproaldehyde, nonanal, phenylacetaldehyde, glyoxal, Asn, Gln, ethanol, 1-propanol, 1-butanol, isobutanol, isoamyl alcohol, 3-methylbutanol, phenethyl alcohol, acetic acid, lactic acid, caproic acid, ethyl acetate, isopentyl acetate, ethyl caproate, ethyl lactate MES and TMP (weighed 1 g for each and 2.5 g for glyoxal) were dissolved in ethanol (10 mL) to obtain 100 g/L stock solution and stored at −20 °C. These stock solutions were diluted with distilled water to obtain 15 mg/L of each. These analytes (2 μL) were reacted with FPN1 solution (50 μL acid solution, 148 μL probe solution) for 20 min to record fluorescence intensity.

### 2.6. Fluorescence Sensing Method Base on FPN1

In order to perform the measurement, the detection procedure was as follows: the acid solution shown in 2.4 (50 μL for reaching the optimum pH 2.1), the FPN1 solution (148 μL) and the sample solution (2 μL) were added into a 96-well microtiter plate. After waiting for 20 min at temperature of 5 °C, the fluorescence intensity of the mixtures was recorded on the fluorescence spectrometer (excitation wavelength: 485 nm; emission wavelength: 553 nm; slit width: 2 nm; temperature: 25 °C.).

The linear regression equation was as follows:F_553nm_ = 344.05C_s_ + 12.82 (R^2^ = 0.9992)(1)
in which C_s_ is the concentration of the acetaldehyde in samples, and the unit of C_s_ is mg/L.

### 2.7. Background Removal Method (BRM)

For the purpose of removing interference of pyruvate and acetal, we proposed an effective pretreatment called BRM.

In order to identify the optimal concentration multiple during distillation, the relationship between the concentration multiple and the recovery of acetaldehyde was also tested. The concentration multiple was obtained by changing the volume of the distillate sample solution.

The BRM procedure was as follows: one drop of defoamer (as a surfactant to prevent beer from spilling during distillation) was added to the samples of beer. The samples of liquor were diluted by 50 times with distilled water. The samples of *Huangjiu* and wine were diluted by 25 times with distilled water. After preheating the steam generator, each sample (100 mL) was poured into the distillation chamber and distilled to remove the background interference. Finally, the distillate sample solution (20 mL) was obtained within 2 min.

The process of the fluorescence sensing method based on FPN1 with BRM can be seen more clearly in Figure 2. As described in 2.6, the concentration (C_s_) of the distillate sample solution could be measured. In order to calculate the original concentration of acetaldehyde in alcoholic beverages, the concentration (C_s_) of the distillate sample is added into the following formula:C = C_s_/N,(2)
in which C is the original concentration of acetaldehyde in the sample of alcoholic beverages, the unit of C is mg/L and N is the actual concentration multiple: N = 5 for beer, N = 0.1 for liquor and N = 0.2 for *Huangjiu* or wine.

### 2.8. Anti-Interference Analysis

In order to reflect real sample situations and to show the response value of acetaldehyde towards FPN1 in the presence of other interferences, the anti-interference analysis was carried out using model samples. According to the reported concentrations of carbonyls, alcohols, esters, acids and amino acids in beer, liquor, *Huangjiu* and wine [2,3,4,5,6,7,8], the average concentrations of acetaldehyde and interference analytes in beer, liquor, *Huangjiu* and wine are shown in Appendix A.

By weighing 1g for each analyte showed in Appendix A and dissolving it in ethanol (10 mL), 100 g/L stock solution was obtained (stored at −20 °C). By diluting and mixing the stock solution, a single standard solution and a mixed standard solution was obtained for each model sample, according to the concentration shown in Appendix A.

The anti-interference analysis without BRM was conducted as follows: the fluorescence intensity was recorded by reacting the standard solution (2 μL) and the mixed standard solution (2 μL), respectively, with the FPN1 solution (50 μL acid solution, 148 μL probe solution) for 20 min.

The anti-interference analysis with BRM was conducted as follows: the standard solution and mixed standard solution were processed through BRM to obtain the distillate sample solution, and the fluorescence intensity and fluorescence spectra were recorded by reacting the distillate sample solution (2 μL) with the FPN1 solution (50 μL acid solution, 148 μL probe solution) for 20 min.

### 2.9. Samples of Alcoholic Beverages

The 3 samples of beer (lager with different original gravity), 6 samples of liquor (different flavor), 6 samples of *Huangjiu* (different types of sweetness) and 6 samples of wine (red wine with different types of sweetness) were purchased from the local supermarket (Wuxi, China). The 3 samples of beer fermentation broths were produced by our own laboratory based on the reported studies [1]. The basic information of each sample is showed in Appendix A.

Model samples (shown in Appendix A) and some randomly selected real samples (sample number 3 of beer and liquor, sample number 5 of *Huangjiu* and wine shown in Appendix A) were used to verify the recovery. Different volumes (50 μL, 100 μL, 200 μL for beer, 500 μL, 1000 μL, 2000 μL for liquor, 250 μL, 500 μL, 1000 μL for *Huangjiu* and wine) of acetaldehyde standard solution (1 g/L) were added into each sample (100 mL) of the model solution, beer, liquor, *Huangjiu* and wine. The concentration of acetaldehyde before and after standard solution addition in each sample was measured by the fluorescence sensing method with BRM. Then, the recovery of acetaldehyde measurement in both model solutions and real samples was calculated.

### 2.10. GC Method

In order to compare the fluorescence sensing method with the common method for detecting acetaldehyde [28], the GC method was performed as follows: 4 mL of samples, 1 mL of internal standard solution (3-heptanone, 30 mg/L) and 1.5 g of NaCl were added to a 15 mL vial and sealed with a silicone septum cap. The HS-GC analysis was conducted on a chromatography column DB-23 (60.0 m × 0.32 mm × 0.25 m). It was set at 200 °C for the injection port and 250 °C for the detector. The column oven temperature was initially held at 40 °C for 8 min, then increased to 180 °C at a steady rate of 10 °C/min. N_2_ (99.99% pure) was used as a carrier gas, and flowed at a rate of 30 mL/min. The total chromatographic run time was 25 min. The LOD was 0.01 mg/L.

### 2.11. Statistical Analysis

The experimental results were expressed as the mean ± standard deviation, and error bars were shown on the graphs. The sensing mechanism of FPN1 to acetaldehyde was measured based on Gaussian 09W.The significant difference between the acetaldehyde concentration in alcoholic beverages by different methods, as well as the fluorescence intensity in different solvents, were evaluated by the Tukey post hoc test. Statistically significant differences were considered for *p* < 0.05, with a 95% confidence level. These statistical analyses were performed using SPSS version 22. The consistency test was measured based on Blant–Altaman [29].

## 3. Results

### 3.1. Design and Synthesis of FPN1

As a kind of common fluorescent group, benzoxadiazole can be designed to produce different emission characteristics by changing the substituents at the fourth and seventh positions [30,31]. According to PET, FPN1 was designed to detect acetaldehyde concentration in alcoholic beverages.

As shown in Figure 1, the synthesis route of FPN1 is a common nucleophilic reaction, and the yield of FPN1 could reach 91.2% without further purification. This shows the advantages of mild reaction conditions, a simple synthesis route and simple post-processing. The structure of FPN1 was confirmed using HPLC, HRMS, ^1^H NMR and IR (Figure 3).

### 3.2. Fluorescence Response of FPN1 towards Acetaldehyde

As shown in Figure 4a, the fluorescence intensity of FPN1 with and without acetaldehyde at different pH values, from 1 to 12, was investigated. The results showed that the optimal pH was 2.1. According to previous studies [32], it is easy to determine the fluorescence intensity of FPN1, as carbonyl compounds can reach their highest levels under acidic conditions due to the pH sensitivity. The greater electronegativity at the para position to the hydrazine moiety would enhance the fluorescence intensity of the hydrazone at a pH below 4. Therefore, the acid solution (shown in 2.4 was used to develop the detection method in this work.

In order to improve the effectiveness of the detection method, factors including solvent and the concentration of FPN1 were assessed. FPN1 and acetaldehyde (10 mg/L) were reacted in DMSO, acetonitrile, ethanol, DMF, PBS and water. According to the results shown in Figure 4b, it was found that the fluorescence intensity was the highest when acetonitrile was used as the solvent, and the fluorescence intensity obtained by using the other four solvents was less than 50% of that obtained using acetonitrile. Acetaldehyde was placed in acetonitrile solution at different levels of concentration of FPN1, and it is shown in the results that the optimal concentration of FPN1 was 300 mg/L (Figure 4c).

The fluorescence spectral of FPN1 to different concentrations of acetaldehyde was verified (Figure 4d). Based on the results, it can be concluded that the fluorescence intensity at 553 nm gradually increased. Specifically, the fluorescence could be clearly seen under the ultraviolet, which indicates that FPN1 has the potential to be developed as a visual sensor.

The fluorescence intensity of FPN1 to acetaldehyde was recorded in acetonitrile with and without light, in order to indicate the light sensitivity of the probe (Figure 4e). According to the results, the presence or absence of light makes little difference in the detection method, which demonstrates the good photostability of FPN1.

The time–response study of FPN1 on different concentrations of acetaldehyde was investigated in acetonitrile (Figure 4f). The results indicate that the fluorescence intensity reaches equilibrium at 20 min. Importantly, the reaction time of the detection method proposed in this study is more rapid than other fluorescent sensing methods described previously [17,23], which indicates the advantage of a quick response towards acetaldehyde.

### 3.3. Analysis of the Selectivity

The selectivity of FPN1 was confirmed by reacting FPN1 with 39 analytes. As the results show in Figure 5, the fluorescence intensity of acetaldehyde is the highest compared with the other analytes. Because the central carbon atoms of some microgram-level carbonyl compounds in beer are highly active, such as propionaldehyde, butyraldehyde and isobutyraldehyde, their carbonyl group will also efficiently compete for the electronics provided by -NH-NH_2_, resulting in a fluorescence intensity of 30–50% of acetaldehyde. On the other hand, the fluorescence intensity of milligram-level carbonyl compounds in beer is 3.5–7.5% of acetaldehyde. Moreover, the fluorescence intensity of alcohol and ester compounds, which are the main flavor substances in liquor, *Huangjiu* and wine, is much lower.

Both Asn and Gln contain the carbonyl group (C=O), which is different from other amino acids and may cause the interference, so the selectivity of FPN1 to Asn and Gln was also assessed. The results show that the fluorescence intensities of Asn and Gln are 0.2% and 1.9% of acetaldehyde, respectively. It is also worth considering the selectivity of FPN1 to acetal and pyruvate, which reach a relatively high concentration in alcoholic beverages. Based on the results, the fluorescence intensities of acetal and pyruvate are 28.4% and 9.4% of acetaldehyde, respectively.

### 3.4. The Anti-Interference Analysis

The variety of flavor components in alcoholic beverages is complex, and the concentration of each flavor component differs between them. For instance, over 60% of aldehyde in beer is acetaldehyde, and over 90% of aldehyde in wine is acetaldehyde. Additionally, it is also worth noting the concentration of pyruvate is 3–10 times higher than that of acetaldehyde in beer, and the concentration of acetal is 2–3 times higher than that of acetaldehyde in beer and liquor (as shown in Appendix A). Therefore, in order to show the real situation of the samples of alcoholic beverages, it is necessary to simulate the actual concentration of anti-interference analytes. There are also several pieces of research which have confirmed the anti-interference by simulating the actual concentration of analytes [33,34,35,36], which allows for a better understanding of practical application for the fluorescence sensing method.

In order to identify the optimal concentration multiple during distillation, the relationship between the concentration multiple and the recovery of acetaldehyde was investigated (Appendix A). The results show that the recovery could be close to 100% using both distillation apparatus 1 and distillation apparatus 2 when the concentration multiple is 5. Importantly, distillation apparatus 1 was applied for the fluorescence sensing method, because distillation 1 (each distillation took 2 min) showed better performance with regard to effectiveness compared with distillation apparatus 2 (each distillation took 5 min).

Anti-interference analysis was examined with (Appendix A) and without (Appendix A) BRM. As the results show in Appendix A, theoretically, the detection value of acetaldehyde in beer might be about 30% higher in the presence of 100 mg/L pyruvate, and the detection value of acetaldehyde in liquor might be about 25% higher in the presence of 600 mg/L acetal. These findings suggest that the interference of pyruvate and acetal is strong before distillation, which especially affects the detection of acetaldehyde in beer and liquor.

Nevertheless, according to the fluorescence spectral of the anti-interference analysis after distillation based on BRM (Figure 6), we could observe that only acetaldehyde obviously shows a strong fluorescence intensity at 553 nm. The anti-interference analysis revealed that BRM could be used to remove the interference of pyruvate and acetal, which would provide a solution to directly combine FPN1 with acetaldehyde in order to enhance the sensitivity, especially for beer and liquor.

### 3.5. Sensing Mechanism of FPN1 towards Acetaldehyde

As shown in Appendix A, the Job’s Plot method was used to explore the binding stoichiometry between FPN1 and acetaldehyde. In order to further understand the response mechanism of FPN1 and acetaldehyde, using the B3LYP method of Gaussian 09W program package, density functional theory (DFT) calculation was performed at the basis set level of 6-31++G(d). The molecular properties of FPN1 and FPN1-CH_3_CHO were calculated. The optimized geometric configuration, the highest occupied orbital (HOMO) and the lowest occupied orbital (LUMO) profiles of FPN1 and FPN1-CH_3_CHO are shown in Appendix A. The energy ranges between the HOMO and LUMO energy levels of FPN1 and FPN1-CH_3_CHO were 0.11288 ev and 0.09605 ev. Compared with FPN1, the energy gap of FPN1-CH_3_CHO was relatively low, resulting in a red shift in the fluorescence spectrum. Using time-dependent density functional theory (TD-DFT) theoretical calculations, the HOMO and LUMO of FPN1-CH_3_CHO at the ground state and excited state were further studied, which is shown in Appendix A. The maximum absorption peaks are 482 nm (f = 0.3502) and 583 nm (f = 0.3598), which is consistent with the actual detection results. Therefore, all results indicate that PET is inhibited and the nucleophile substitution reaction occurs.

### 3.6. The Validity of the Fluorescence Sensing Method with BRM

In order to further distinguish the performance of the fluorescent sensing method with BRM, the validity of the method was identified. As shown in Table 1, the recovery was from 94.02% to 108.12%, which proves the accuracy of the quantified acetaldehyde concentrations by the fluorescence sensing method with BRM. The average RSD was 2.08%, which shows the high stability of this method.

The fluorescence intensity of FPN1 was linear with the concentration of acetaldehyde in the range of 0.0053–200 mg/L (Appendix A; R^2^ = 0.9992). LOD was 0.0016 mg/L based on Cim = 3SD/B (SD represents standard deviation of the blank solution and B is the slope of the linear regression), according to the definition of the International Union of Pure and Applied Chemistry (IUPAC) [19]. As shown in Appendix A, compared with the reported sensing methods [17,23], the linear range of this method is wider and the LOD is lower. Although the advantage in reaction time is not obvious, this method showed the advantage of lower cost of reagents and materials.

Remarkably, due to FPN1 combining with acetaldehyde directly, the linear range and LOD of the fluorescent sensing method proposed in this study showed a much better performance than the previous GC methods (Appendix A) [7,28,37,38,39,40].

### 3.7. Detection of Acetaldehyde in Alcoholic Beverages by the Fluorescence Sensing Method and GC Method

The acetaldehyde concentration in the samples of beer, liquor, *Huangjiu* and wine was determined by the GC method and the fluorescence sensing method with or without BRM. All results are shown in Figure 7.

The concentration of acetaldehyde in finished beer (sample number 1–3 of beer) is generally below 10 mg/L, and lower than that in fermentation broth (sample number 4–6 of beer), which is about 10–35 mg/L, due to the process of high concentration dilution. The concentration of acetaldehyde in *Huangjiu* (10–250 mg/L) and wine (5–200 mg/L) is higher than that in beer, due to the different production process of saccharification and fermentation. The concentration range of acetaldehyde in liquor is about 200–1000 mg/L, which is much higher than other alcoholic beverages, due to the process of distillation.

According to the significant difference analysis shown in Figure 7, there is no statistically significant difference between the results of the fluorescence sensing method with the BRM and GC method (*p* = 0.89 > 0.05). In order to further indicate that the fluorescence sensing method with BRM could be an alternative detection method to the GC method, a consistency test was undertaken by Bland–Altman, and the results are shown in Appendix A. It was found that all the differences are between the upper and lower 95% limits of agreement, which indicates that the fluorescence sensing method is in good agreement with the BRM and GC method.

As expected, the results of the fluorescence sensing method without the BRM and GC method show statistically significant differences (*p* = 0.043 < 0.05). Especially for the samples of beer and liquor, the results are significantly higher than those obtained with GC, which verifies that pyruvate and acetals affect the detection results.

More importantly, the fluorescence sensing method with BRM was more rapid than with the GC method. The average chromatographic run time of each sample by the GC method was about 30 min (approximately 32 h for 24 samples with 3 times parallel). For the fluorescence sensing method with BRM, the distillation time and the reaction time for each sample were 2 min and 20 min, respectively, which involves an analysis time for a single sample which is similar to the GC method. Benefiting from the application of a fluorescence detector and the concurrent detection of a batch of samples, all of the samples could be reacted in a 96-well plate concurrently in order to improve the detection efficiency (approximately 1.2 h for 24 samples with 3 times parallel). Furthermore, comparing it to other preprocessing methods of GC reported recently, BRM is faster (Appendix A). In summary, it can be clearly found that the fluorescence sensing method with BRM could play an important role in improving the detection efficiency.

Considering that the operation of distillation in BRM may lead to an intensive procedure, GC methods still need be used in the detection of large number of samples. We may focus on the optimization of pretreatment and the development of specific probes in the future, which remains a daunting challenge.

The fluorescent sensing method based on FPN1 has greatly improved the ability of effectiveness while also ensuring accuracy, and has a set of advantages: it is low-cost, rapid and requires no complicated equipment compared to prior methods. Therefore, the fluorescent sensing method based on FPN1 with BRM is well-suited for the detection of acetaldehyde in alcoholic beverages, especially for small-scale samples.

## 4. Conclusions

In summary, a novel fluorescence sensing method for acetaldehyde in alcoholic beverages based on FPN1 was developed. The function of FPN1 relies on the nucleophile substitution reaction and PET. As a result of the simple synthetic route, as well as its high purity, FPN1 can be prepared in large quantities. FPN1 could completely react with acetaldehyde in 20 min under acidic conditions, with a maximum excitation wavelength of 553 nm. More importantly, BRM, which we proposed in this study, has greatly improved the ability to remove the interference, and is of great significance for practical application. The results indicate that the fluorescence sensing method with BRM has good linearity in the range of acetaldehyde concentration of 0.0053–200 mg/L. In addition, the LOD is 0.0016 mg/L, which shows a wider range and more sensitive performance than prior methods. Meanwhile, the determined concentrations were comparable to those measured by GC method, which validates the accuracy of the fluorescence sensing method with BRM and demonstrates the advantage of low cost and rapidity, particularly for small-scale samples. The fluorescence sensing method, based on FPN1, presents a novel, rapid and sensitive detection strategy for the beverage industry.

## Figures and Tables

**Figure 1 foods-11-03450-f001:**
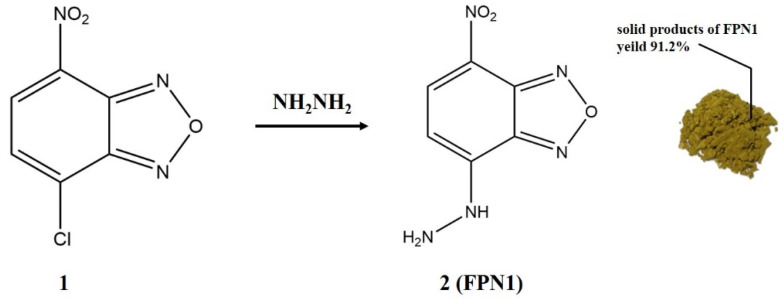
Synthesis scheme and morphology of FPN1.

**Figure 2 foods-11-03450-f002:**
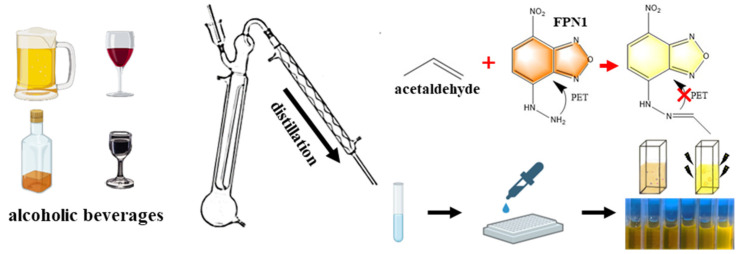
The schematic diagram of the fluorescence sensing method based on FPN1 with BRM.

**Figure 3 foods-11-03450-f003:**
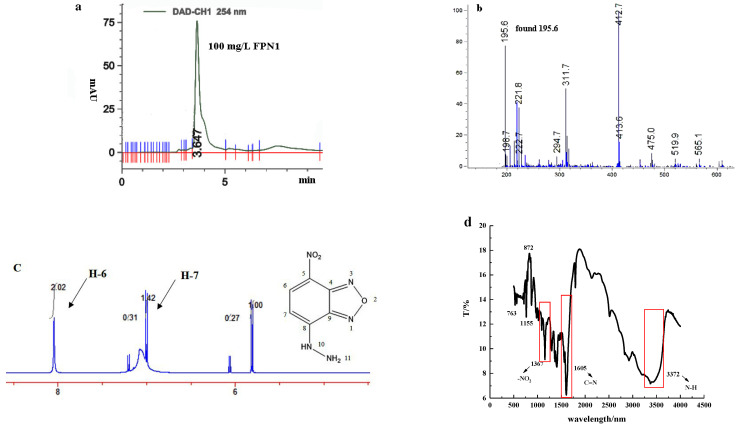
(**a**) HPLC spectrum of FPN1. (**b**) HRMS spectrum of FPN1. (**c**) ^1^H NMR spectrum of FPN1. (**d**) IR spectrum of FPN1. ^1^H NMR (300 MHz, DMSO), δ 8.08 (ppm): (d, J = 10.8 Hz, 2H), 7.02 (d, J = 10.8 Hz, 1H). HRMS (ESI): calculated for [M + H]+ 195.04, found 195.6. All the key groups (-NO_2_, C=N and -NH) could be observed on the IR spectrum.

**Figure 4 foods-11-03450-f004:**
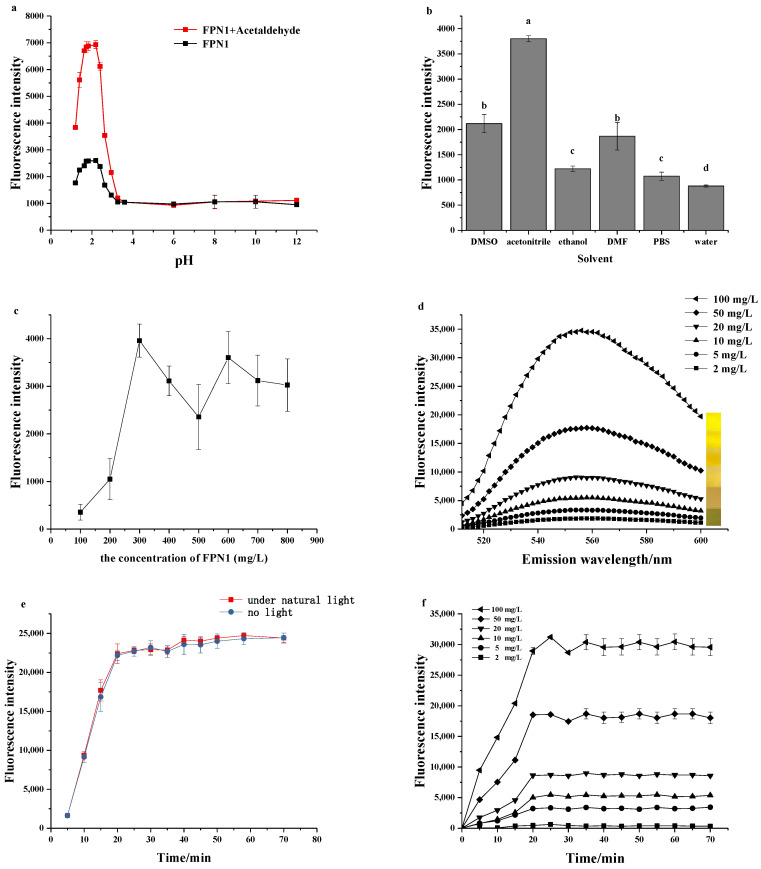
(**a**) Fluorescence intensity of FPN1 (300 mg/L) with and without acetaldehyde (10 mg/L) at different pH (1–12). (**b**) Fluorescence intensity of FPN1 (300 mg/L) in the presence of acetaldehyde (10 mg/L) in different solvents. Statistically significant differences were considered for *p* < 0.05. ^a,b,c,d^: Values followed by the same letter do not differ significantly. (**c**) Fluorescence intensity of acetaldehyde (10 mg/L) in the presence of different concentrations of FPN1. (**d**) Fluorescence spectra of FPN1 (300 mg/L) in the presence of acetaldehyde (2, 5, 10, 20, 50, 100 mg/L). (**e**) Time-dependent fluorescence intensity changes of FPN1 (300 mg/L) in the presence of acetaldehyde (100 mg/L) with or without light. (**f**) Time-dependent fluorescence intensity changes in FPN1 (300 mg/L) in the presence of acetaldehyde (2, 5, 10, 20, 50, 100 mg/L). All tests were performed in triplicate.

**Figure 5 foods-11-03450-f005:**
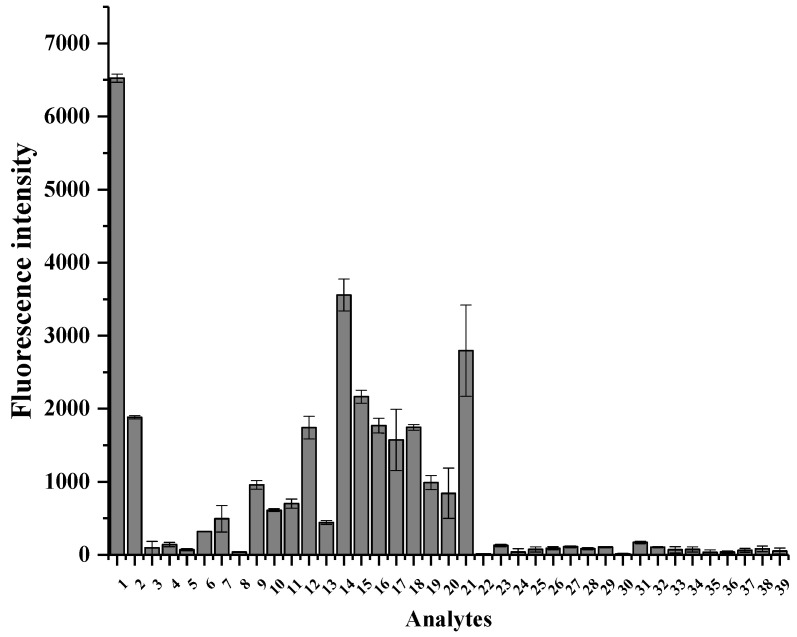
Fluorescence intensity of FPN1 (300 mg/L) upon addition of various analytes (15 mg/L for each, 1, acetaldehyde; 2, acetal; 3, furfural; 4, 5-HMF; 5, acetoin; 6, 2,3-butanedione; 7, 2,3-pentanedione; 8, 2.3-hexanedione; 9, pruvate; 10, propanone; 11, hydroxypropanone; 12, α-ketoglntaric acid; 13, methylglyoxal; 14, propionaldehyde; 15, butyraldehyde; 16, isobutyraldehyde; 17, isovaleraldehyde; 18, caproaldehyde; 19, nonanal; 20, phenylacetaldehyde; 21, glyoxal; 22, Asn; 23, Gln; 24, ethanol; 25, 1-propanol; 26, 1-butanol; 27, isobutanol; 28, isoamyl alcohol; 29, phenethyl alcohol; 30, 3-methylbutanol; 31, acetic acid; 32, lactic acid; 33, ethyl acetate; 34, isopentyl acetate; 35, caproic acid; 36, ethyl caproate; 37, ethyl lactate; 38, MES; 39, TMP. The test was performed in triplicate.

**Figure 6 foods-11-03450-f006:**
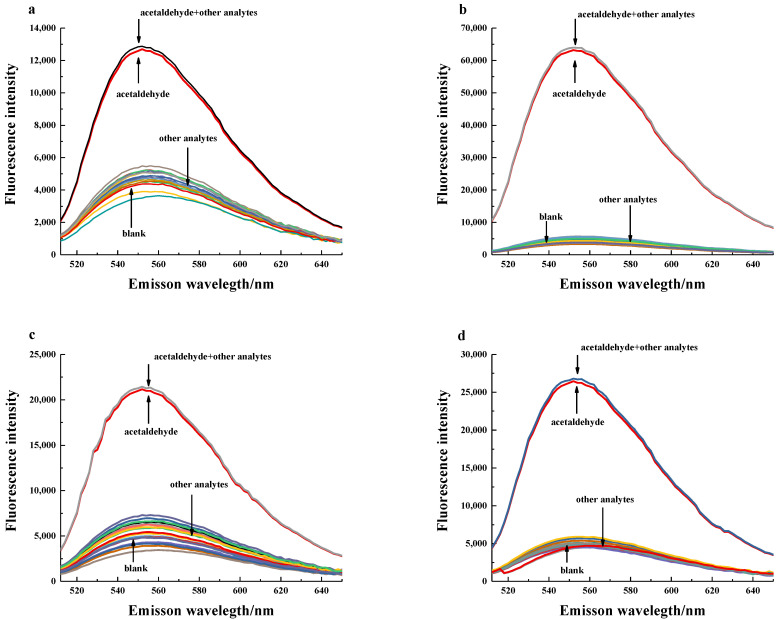
(**a**) Fluorescence spectrum of anti-interference analysis of beer with BRM. (**b**) Fluorescence spectrum of anti-interference analysis of liquor with BRM. (**c**) Fluorescence spectrum of anti-interference analysis of *Huangjiu* with BRM. (**d**) Fluorescence spectrum of anti-interference analysis of wine with BRM. The concentration of acetaldehyde and other analytes are shown in Appendix A.

**Figure 7 foods-11-03450-f007:**
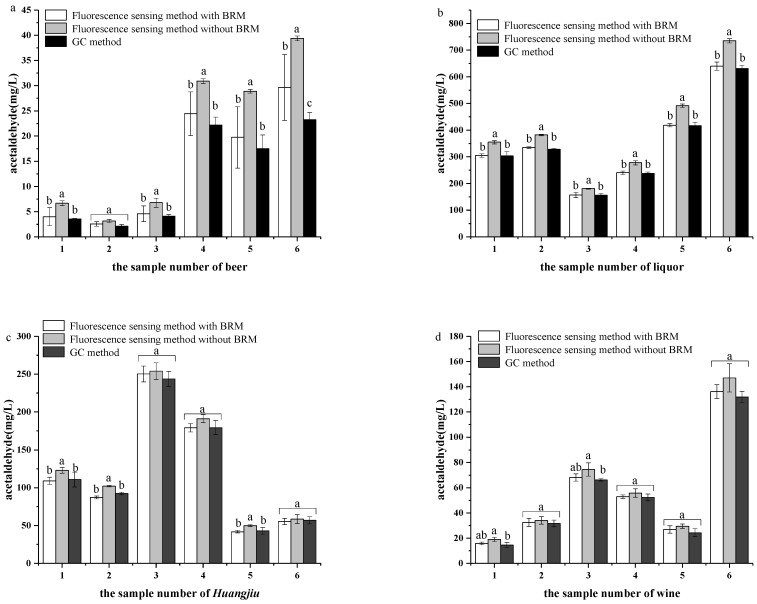
(**a**) Determination of acetaldehyde concentrations in the samples of beer by the fluorescence sensing method and GC method. (**b**) Determination of acetaldehyde concentrations in the samples of liquor by the fluorescence sensing method and GC method. (**c**) Determination of acetaldehyde concentrations in the samples of *Huangjiu* by the fluorescence sensing method and GC method. (**d**) Determination of acetaldehyde concentrations in the samples of wine by the fluorescence sensing method and GC method. Mean values in each detection method group with superscript letters indicate significant differences. Statistically significant differences were considered for *p* < 0.05. ^a,b^: Values followed by the same letter do not differ significantly. The basic information of each sample is shown in Appendix A. Tests were performed in triplicate.

**Table 1 foods-11-03450-t001:** The recovery and RSD of the fluorescence sensing method with BRM.

Sample	Initial Acetaldehyde Concentration (mg/L)	Acetaldehyde Spiked (mg/L)	Acetaldehyde Concentration Detected (mg/L)	Recovery (%)	RSD%
Model beer	10.00	1	10.95 ± 0.42	95.36	3.83
	10.00	2	11.99 ± 0.24	99.75	2.00
	10.00	4	13.76 ± 0.36	94.87	2.61
beer	5.25	1	6.43 ± 0.15	102.80	1.43
	5.25	2	7.08 ± 0.21	97.66	1.83
	5.25	4	9.01 ± 0.09	97.40	0.67
Model liquor	300	10	313.03 ± 4.21	101.97	3.22
	300	20	316.47 ± 2.45	98.90	2.45
	300	40	344.48 ± 3.68	101.31	3.87
Liquor	122.09	10	129.49 ± 1.88	98.03	1.45
	122.09	20	153.63 ± 2.04	108.12	1.32
	122.09	40	163.81 ± 2.12	101.06	1.29
Model Huangjiu	50	5	54.19 ± 2.10	98.52	3.45
	50	10	62.40 ± 0.96	104.00	3.46
	50	20	73.38 ± 1.72	104.82	2.14
Huangjiu	49.17	5	51.11 ± 1.65	94.36	3.23
	49.17	10	55.90 ± 1.22	94.47	2.18
	49.17	20	66.49 ± 1.41	96.12	2.12
Model wine	100	5	107.73 ± 1.94	102.60	1.54
	100	10	113.92 ± 2.34	103.6	3.45
	100	20	116.12 ± 3.21	96.77	2.31
Wine	32.44	5	38.81 ± 0.66	103.65	1.70
	32.44	10	41.82 ± 1.53	98.55	3.66
	32.44	20	49.30 ± 0.92	94.02	1.86

## Data Availability

Data are contained within the article or supplementary material. The data presented in the figures of this study are available upon request from the corresponding author.

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
