# Peer review of "Development of a Rapid and Sensitive Fluorescence Sensing Method for the Detection of Acetaldehyde in Alcoholic Beverages"

_foods, 2022, doi:10.3390/foods11213450_

Round 1

Reviewer 1 Report

Article: Development of a rapid and sensitive fluorescence sensing method for the detection of acetaldehyde in alcoholic beverages

The manuscript describes the development of a fluorescent sensor for acetaldehyde detection in alcoholic beverages. Despite the interesting results, the presentation of the work should be thoroughly improved.

Abstract: “With the development of fluorescence methods, fluorescent probe could be used as a new tool for the 10 detection of acetaldehyde”. I think that this sentence is suitable in abstract. Indeed, the authors should mention fluorescence advantages for which this detection method was selected.

What do the authors mean by GC method? It is not clear! Please clarify

I think that the abstract should be rewritten; the work principle should be highlighted in this part.

Introduction:

Line 27: Please explain these concentrations? Are they maximum permissible amounts of acetaldehydes in alcoholic beverages? If not, the maximum amounts should be mentioned!

Line 53: “Therefore, here we developed a fluorescence sensing method based on fluorescent probe N1 (FPN1)”. This sentence is not enough to describe the objective and principle of the reported work.

Introduction part should describe the objective of this study, novelty of the method, principle and main results as well as advantages of the proposed method as compared to the existing ones in the literature. More details about FPN1 should be given.

Highlights should be removed from introduction part.

Materials and methods:

Figure 1.c. the steps should be explained in the caption or in the text.

Line 170: please define the acid solution used here!

Results and discussion:

Line 238: please explain the effect of pH on the fluorescence intensity and discuss the obtained result.

Please change the expression “the results are showed” by have shown in all the text.

The selectivity study should be discussed deeply by explaining the high fluorescence intensities obtained with some compounds.

Please explain why the samples are spiked with acetaldehyde while all the beverages tested here already contain a known concentration of acetaldehyde?

Line 388: please compare the performance of your method with other sensing techniques reported in the literature.

Please check the figure numbers

Reviewer 2 Report

Manuscript: Foods (Fluorescence Sensing)

This manuscript describes the development of a fluorescence sensing method for detection and quantification of acetaldehyde in different alcoholic beverages (beer, liquor, Huangjiu and wine). There is undoubtedly a need for rapid analytical methods that overcome issues associated with traditional quantitation using GC-MS and LC-MS (in particular, time-consuming sample preparation, run times and data analysis). As such, this paper reports a worthy study. However, there are some issues that need to be addressed.

There are some issues with language expression (plural vs singular – e.g. probe vs probes, researches vs research; clumsy wording – e.g. ‘which was listed as a group 1 carcinogen’, FPN1 solution was consist of…’, ‘Each sample was distillated…’; and overly long sentences – e.g. L27 – 32, but also long lists of compound names). Additionally there are typos that more careful proof-reading should have captured – e.g. repeated use of ‘simple’ instead of ‘sample’ (e.g. L185 and L187, and elsewhere), and ‘blinding’ instead of ‘binding’ (L357). This is by no means an exhaustive list, but provides some examples of issues that should be addressed to improve the readability/quality of expression, prior to publication.

Keywords – these are all used in the title – are there not other keywords that could be included?

Introduction

What is the origin of acetaldehyde in alcoholic beverages? Presumably fermentation? This seems like an important aspect of background information that should be included in the Introduction. Also, what do beverage producers do to mitigate and/or manage acetaldehyde concentrations?

M&M

My key concern with this paper is the lack of detail with much of the M&M.

Section 2.1 lists the chemicals/solvents used. Glyoxal was only 40% pure – was it purified prior to use? The instruments listed in this section simply have their country of origin provided at the end, and not the manufacturer name and address, e.g. as was done for Sigma Aldrich in the first paragraph.

Section 2.2 describes the synthesis of FPN1 (a known compound). The M&M should include characterisation data (e.g. accurate mass, NMR etc.); it should not be buried in the Supplementary Materials (in Figures that are too small to easily read/interpret) and with reference to published characterisation data. Yield is reported twice in L109, but a percentage yield is not given. The size of Figure 1 should be increased to make it easier to read – but Figure 1b adds nothing, and Figure 1c should be presented separately, where it is cited in section 2.8. Additionally, the image shows a beaker with samples in a pouring position, which might confuse some readers who interpret this as pouring the sample into the distillation glassware, whereas presumably the sample is in the round bottom flask at the start of distillation? This should be removed. Again, the image size should be increased.

Section 2.4 refers to 1 g of each analyte being dissolved in 10 mL of water to obtain a 10 g/L stock solution, but 1 g in 10 mL is 100 g/L. Why were stock solutions made up in water and not ethanol? Are there any solubility issues? Especially at such high concentrations? This section describes the preparation of analytes for selectivity analyses, but the actual selectivity analyses (i.e., what was done with these samples) is not described.

Section 2.5 refers to anti-interference analysis and seemingly uses standards/analytes described in section 2.4, but again, the nature of anti-interference analyses is not actually described. These analyses could not be replicated by others based on the information reported in the M&M, thus it is not adequately described and must be revised. Section 2.5 refers to ‘simulated samples’ (L134) but there is no explanation of what these are or how they were prepared. I would expect in analytical method development that model solutions (e.g. model beer, model wine, model spirit) spiked with analytes of known concentration would be used to test reproducibility, sensitivity, precision, recovery, etc. but if this has been done, it hasn’t been adequately described (and if it hasn’t been done, it should have been). Standard development and validation of quantitative methods requires limits of detection, linear ranges, instrument repeatability, recovery and reproducibility to all be reported for the optimised conditions. Recovery is described in section 2.6 for real samples – but at an addition rate of 1 g/L – which is significantly higher than the 10s to 100s of mg/L acetaldehyde concentrations given for the different alcoholic beverages in the Introduction. Furthermore, Table 1 suggests different rates of addition of acetaldehyde to different beverages, with BRM, and with six replicates. This is not consistent with what is reported in the M&M at all.

L138 to L160 could be better presented in table format.

Additionally, section 3.2 of the R&D refers to optimisation work (effect of pH, solvent, FPN1 concentration, emission wavelength, reaction time – section 3.2) that is not described in the M&M.

Section 2.6 explains that commercial samples of alcoholic beverages were sources from a supermarket. However, it’s not clear how many of each sample were purchased and analysed. On what basis were these alcoholic beverages chosen; e.g. price? style? region of origin? popularity in market? Reference is also made to beer fermentation broths made in-house, but no other details are provided.

Section 2.7 describes the fluorescence sensing method, and refers to the use of distillate sample solutions, but besides reference to distillation in the Introduction, there has been no prior explanation of the distillation process/M&M.

Section 2.8 refers to a defoamer (L181), but the nature/origin of this material has not been provided. Similarly, reference is made to a collection buffer (L184), but the nature/origin of this material is not explained.

Section 2.8 refers to samples being distilled to remove ‘background interference’, but it is not clear what the interfering compounds might be. More importantly, there is no explanation of how distillation is performed. The Supplementary Materials (Figure S2) includes two diagrams of distillation equipment and an ‘optimal concentration ratio’ figure, but there is no M&M describing distillation trials. Equation 2 includes a concentration multiple, but it is not clear how the different values of N (L190-191) are determined for the different alcoholic beverages. Figure 1c is cited here, but this provides no adequate explanation of the distillation process. Critically, the R&D suggests distillations of 100 mL of alcoholic beverages were performed in 2 or 5 min (L317–318). I cannot reconcile this with my own experience of laboratory-scale distillation using glassware shown in Figure S2a and S2b; the time required to heat 100 mL of sample to achieve vaporisation of sample would surely exceed 2 min, let alone 5 min, so this seems implausible to me. The distillation M&M must be provided in more detail. This is a critical component of method development and it is barely described, let alone reported in sufficient detail so as to enable replication by others.

Section 2.9 describes quantitation of acetaldehyde using traditional GC analysis If this is a common method (L185), is there not an appropriate citation? There are several instances of ‘ml’ instead of ‘mL’ in this sentence. The limits of detection for this method should also be stated. If nitrogen was the carrier gas, why is hydrogen flow rate also given?

The authors indicate that the standard GC method for acetaldehyde quantification is 25 min (with no apparent sample preparation aside from the addition of internal standard and sodium chloride – albeit, alcoholic beverages are often diluted to overcome issues with high ethanol concentration, so this seems unusual). Nevertheless, the GC method is 25 min, whereas the fluorescence sensing method involves dilution/pH adjustment and distillation, and then a 20 min reaction time, so presumably involves a similar analysis time to the GC method, which does not involve solvents. So, on what basis are the authors inferring that their method is ‘rapid’? If the efficiency gain relates to the concurrent reaction time for a batch of samples, this needs to be explained.

R&D

Figures are often too small to easily read/interpret.

It is difficult to interpret method development/optimisation work when this hasn’t been adequately described in the M&M.

Figure captions (e.g. Figure 2) are too long. This information should be tabulated in the M&M and cited in the Figure caption, not repeated.

There are formatting inconsistencies in Table 1. Also column headings should be revised. Surely column 2 could just be Initial acetaldehyde concentration.

L403-404 explains that samples 1 to 3 in Figure 2a are commercial beers, whereas samples 4 to 6 are fermentation broths, but this is not clear from the M&M or the Figure 2 caption. In the case of wines, were these red, white, sparkling, fortified wines? Had wines gone through secondary fermentation and might this (and/or variety?) explain variation in acetaldehyde concentrations? Similarly for liquors – there is no detail regarding the nature of the liquors and whether this might influence their composition.

How is the analysis time of 1.2 hours for fluorescence sensing calculated? If the efficiency gain relates to concurrent reaction times, then this should be clearly stated here.

How does Figure S1 indicated BRM is much faster (L428) than GC preprocessing methods.

Author contributions – some authors do not appear to meet the criteria for coauthorship. The provision of resources and supervision and/or funding acquisition alone do not warrant coauthorship. Have these authors not even contributed to reviewing/editing the paper?

References

The references are formatted inconsistently.

Round 2

Reviewer 1 Report

the paper can be published in this form

Author Response

It's a great honor to get your recognition of our research!

Reviewer 2 Report

The authors have substantially improved their paper, providing the additional detail in the M&M that was requested and addressing many of the other queries/issues raised. Some of the revised Figures have issues (e.g., Figure 1, Figure 4b are showing black backgrounds, while font sizes are unnecessarily large in Figure 5). 

I would still like to see more detail provided regarding the selection of commercial beer, wine, liquir and Huangjiu samples in section 2.9. It's not a question of whether these samples are 'common' (and this word doesn't address my issue, so should just be removed). It's whether or not it is appropriate to better characterise the nature (i.e. style) of these beverages. For example, are the wine samples red or white wines? Still or sparkling wines? And section 2.9 should specify how many of each sample there is. On what basis were the particular samples chosen? Price? Style? Popularity? Maybe it doesn't matter - in which case the authors could state they were chosen randomly. The lack of detail is the issue.

Finally, there are still some English expression issues (and typographical errors), but presumably the editorial team can address these in production.

Author Response

Response to Reviewer 2 Comments

Point1: Some of the revised Figures have issues (e.g., Figure 1, Figure 4b are showing black backgrounds, while font sizes are unnecessarily large in Figure 5).

Reponse1: Thanks for pointing these issues! We have readjusted the relevant Figures.

Point 2: I would still like to see more detail provided regarding the selection of commercial beer, wine, liquor and Huangjiu samples in section 2.9. It's not a question of whether these samples are 'common' (and this word doesn't address my issue, so should just be removed). It's whether or not it is appropriate to better characterise the nature (i.e. style) of these beverages. For example, are the wine samples red or white wines? Still or sparkling wines? And section 2.9 should specify how many of each sample there is. On what basis were the particular samples chosen? Price? Style? Popularity? Maybe it doesn't matter - in which case the authors could state they were chosen randomly. The lack of detail is the issue.

Reponse2: We really thank you for pointing this issue.

We selected beer samples with different original gravity. All beer samples used in this study are lager (marked in the revised submission and Table S1). Most commercial beers in supermarkets are lager in China, so we only selected lager as the beer samples.

We selected liquor samples with different flavors. In general, flavor is the main classification factor of liquor.

We selected Huangjiu samples with different sugar content. In general, type of sweetness is the main classification factor of Huangjiu. The residual sugar content was also added in Table S3.

We selected wine samples with different sugar content (The residual sugar content was also added in Table S4). However, all wine samples used in this study are red wine (marked in the revised submission). We did not select white wine, still wine or sparkling wine in this study. We may test these samples in future studies, but in this study, we mainly focus on the comparison among different detection methods. We really appreciate you for pointing out the shortcomings of our research.

The samples used for recovery verification are randomly selected (marked in the revised submission).

We have revised related sentences and added more information of the alcoholic beverage samples in lines 208-215:

“The 3 samples of beer (lager with different original gravity), 6 samples of liquor (different flavor), 6 samples of Huangjiu (different types of sweetness) and 6 samples of wine (red wine with different types of sweetness) were purchased from the local supermarket (Wuxi, China). The 3 samples of beer fermentation broths were produced by our own laboratory based on the reported studies. The basic information of each sample is showed in Table S1-S4.

Model samples (shown in Table S5) and some randomly selected real samples (sample number 3 of beer and liquor, sample number 5 of Huangjiu and wine shown in Table S1-S4) were used to verify the recovery.”